# How Population Aging Affects Industrial Structure Upgrading: Evidence from China

**DOI:** 10.3390/ijerph192316093

**Published:** 2022-12-01

**Authors:** Xiao Shen, Jingbo Liang, Jiangning Cao, Zhengwen Wang

**Affiliations:** 1School of Business, Xinyang Normal University, Xinyang 464000, China; 2School of Business Administration, Zhongnan Uninersity of Economics and Law, Wuhan 430073, China; 3School of Economics and Management, Wuhan University, Wuhan 430072, China; 4National Institute of Insurance Development, Wuhan University, Ningbo 315100, China

**Keywords:** macroeconomic environment, population aging, industrial structure upgrading

## Abstract

The question of how to proactively respond to population aging has become a major global issue. As a country with the largest elderly population in the world, China suffers a stronger shock from population aging, which makes it more urgent to transform its industrial and economic development model. Concretely, in the context of the new macroeconomic environment that has undergone profound changes, the shock of population aging makes the traditional industrial structure upgrading model (driven by large-scale factor inputs, imitation innovation and low-cost technological progress, and strong external demand) more unsustainable, and China has an urgent need to transform it to a more sustainable one. Only with an in-depth analysis of the influence mechanism of population aging on the upgrading of industrial structure can we better promote industrial structure upgrading under the impact of population aging. Therefore, six MSVAR models were constructed from each environmental perspective based on data from 1987 to 2021. The probabilities of regime transition figures show that the influencing mechanisms have a clear two-regime feature from any view; specifically, the omnidirectional environmental transition occurs in 2019. A further impulse–response analysis shows that, comparatively speaking, under the new environment regime the acceleration of population aging (1) aggravates the labor shortage, thus narrowing the industrial structure upgrading ranges; (2) has a negative, rather than positive, impact on the capital stock, but leads to a cumulative increase in industrial structure upgrading; (3) forces weaker technological progress, but further leads to a stronger impact on the industrial structure upgrading; (4) forces greater consumption upgrading, which further weakens industrial structure upgrading; (5) narrows rather than expands the upgrading of investment and industrial structures; and (6) narrows the upgrading of export and industrial structures. Therefore, we should collaboratively promote industrial structure upgrading from the supply side relying heavily on independent innovation and talent, and the demand side relying heavily on the upgrading of domestic consumption and exports.

## 1. Introduction

The primary challenge to industrial development is that the macroeconomic environment has undergone profound fundamental changes. On the supply side, the labor force has shifted from being almost unlimited to having a limited surplus and a structural shortage [1]. The high base formed by massive investment has significantly decreased the speed of capital accumulation [2,3]. Furthermore, with the gradual erosion of the advantage of economic backwardness, relying on imitation innovation to achieve low-cost and rapid technological progress has become unsustainable, and that based on independent innovation has slowed down significantly [3]. On the demand side, domestic consumption shows an upward trend in terms of both quantity and quality; that is, the consumption rate has entered the rising stage of the “U-shaped” curve (in terms of industrial and economic development, the change in the consumption rate follows a “U-shaped” pattern of decreasing first, then increasing (Hoffmann et al., 1958; Rostow 1959; Chenery 1989; Kuznets 1971), and consumption levels and structures are growing in intensity, thus playing a stronger role than investment in industrial and economic development. High-speed investment growth is also unsustainable and gradually settling into a downward trend, mainly because of slowing investment in certain important fields such as real estate, infrastructure construction, and manufacturing given the growing resource, environmental, land and labor force constraints [4]. At the same time, consumption investment has been below equilibrium for a long time, thus leading to a significant decline in return on investment. In addition, investment structures are optimizing more rapidly, which has manifested in rapid investment growth in tertiary and high-tech industries and slowing investment growth in energy-intensive industries [5]. There is currently a downturn in external demand, mainly because the world economy faces multiple difficulties and uncertainties as well as uneven recoveries. At the same time, the traditional comparative advantages formed along factor endowments and market forces are gradually being lost [6] and the formation of new high-value comparative advantages is strongly inhibited by non-market factors [7], thus slowing the upgrading of exports. In brief, in the new macroeconomic environment, the existing industrial structure upgrading model cannot be sustained, which is driven by large-scale factor inputs, imitation innovation and low-cost technological progress, investment, and strong external demand.

Population is one of the most important factors affecting industrial structure upgrading [8,9]. In recent years, the aging problem has become more severe: the growth of the proportion of the elderly population is accelerating, the dependency of the population is rising—even crossing the Lewis turning point—and the demographic dividend advantage has been exhausted [10,11,12]. Although the “universal two-child”, “universal three-child” and “delayed retirement” policies were successfully launched, their effect has been limited. In the future, population aging will further accelerate, and therefore continue to shock each aspect of the macroeconomic environment of industrial development.

On the supply side, this trend has a direct and negative impact on the labor supply [13,14,15], thus accelerating the loss of comparative advantages in labor-intensive industries [1] and impeding the growth of service industries. The elderly, who only consume and do not produce, can hardly support the “high savings-high investment” development model [16]. At the same time, capital stock accumulated in earlier stages is very high, and thus is unlikely to continue to grow at the same pace and even suffer negative impact, hardly supporting continuous extensive industrial expansion. The rapidly aging population directly restrains the vitality of technological innovation [17]. At the same time, rising factor endowments and the accumulation of human capital stimulates technological innovation [18]. Additionally, the accelerated disappearance of demographic dividends has a certain mechanism of reversal pressure on technological innovation [19,20]. On the demand side, the accelerated elderly population creates hugely diversified and personalized consumption demands for elderly care [21], while it has different impacts on the consumption of food, transportation, communication, culture, education, and entertainment [22,23,24,25]; the impact on investment led by consumption is also a double-edged sword [18]. The export comparative advantage can therefore be fortified by turning from labor-intensive products to capital- and technology-intensive products [26,27].

As mentioned above, many scholars pay attention to fundamental changes in the macroeconomic environment and its constraints on industrial structure upgrading as well as the impact of population aging on each aspect of industrial development. However, the following issues are often ignored. Is the influencing mechanism of population aging on industrial structure upgrading constrained by the environment? That is, is the influencing mechanism in the new macroeconomic environment different from that in the old economic environment? Is it necessary to comprehensively and systematically investigate the influencing mechanism of population aging on industrial structure upgrading from the perspective of each element of the macroeconomic environment?

In order to answer the above questions, this paper seeks to put forward six theoretical hypotheses from the perspective of each element of the macroeconomic environment (e.g., labor supply, capital, technology, consumption, investment, and exports). Then, a time-varying Markov-Switching Vector Autoregressions (MSVAR) model is used to determine the transition point in the influencing mechanism of population aging on industrial structure upgrading, as well as that of the environment, further to lay special stress on analyzing the influencing mechanism of population aging on industrial structure upgrading in the context of the new macroeconomic environment.

The significance of the research on the aforementioned problems lies in the macroeconomic environment directly restricting industrial structure upgrading, which implies that their influencing mechanisms must be investigated under environmental constraints. In addition, population aging and industrial structure upgrading are two parallel trends in China’s economic and social development, which suggests that industrial structure upgrading must be examined and judged in the context of population aging. In conclusion, this article focuses on the research on the influencing mechanism of population aging on industrial structure upgrading under environmental constraints, and further constitutes a synergy mechanism from the supply and demand sides in adapting to changes in the macroeconomic environment to promote industrial structure upgrading while in market equilibrium. This approach avoids policy side effects such as overcapacity and mismatches between supply and demand, which is significant for China to achieve high-quality growth and become an economic superpower.

## 2. Research Hypothesis and Theoretical Analysis

As shown in Figure 1 [28], population aging shocks the macroeconomic environment, which further affects industrial structure upgrading. The macroeconomic environment includes the supply and the demand sides, which involve such factors as labor supply, capital stock, and technology, as well as consumption, investment, and exports. Furthermore, the influencing mechanism of population aging on industrial structure upgrading is different against different environmental backgrounds. In the past, the macroeconomic environment of industrial development was characterized by an abundant labor supply, rapid capital accumulation, rapid technological progress via imitative innovation, slow upgrading of consumption in the declining stage of the “U-shape”, slow investment upgrading, and rapid upgrading of low-order exports given the factor endowments and market power. In recent years, the macroeconomic environment has undergone fundamental changes, which have been characterized by a short labor supply, slowed capital accumulation, slowed technological progress resorting to independent innovation, rapid upgrading of consumption in the rising stage of the “U-shaped” curve, accelerated optimizing investment structure in the development stage of emphasizing on quality, and slowed upgrading of higher-order exports restricted by more non-market factors. Irrespective of any environmental background, population aging further impacts all aspects of the macroeconomic environment, which affects industrial structure upgrading. This section takes the influencing mechanism of population aging on industrial structure upgrading in the old environment as a comparison and focuses on analyzing that in the new environment.

### 2.1. Labor Supply

In the old macroeconomic environment, there was a relatively young population and an almost-infinite labor force. Even if population aging were to accelerate, its negative impact on the labor supply would be relatively weak. Furthermore, in the past, the manufacturing industry developed faster than the service industry. To be exact, the manufacturing industry developed faster than the service industry in the period of 1981–2011. According to the annual industrial added value and service added value released on the official website of the National Bureau of Statistics in China, we respectively calculated their average annual growth rate. The average annual growth rate of industrial added value from 1981 to 2011 reached 11.5% (we have deflated the data of industrial added value with the industrial added value index based on the year of 1978), much higher than the average annual growth rate of service added value of 6.12% (we have deflated the data of service added value with the service added value index based on the year of 1978). As well, the labor-intensive manufacturing industry developed relatively quickly, which means that the demand for labor factors was relatively high. Therefore, a drop in the labor supply will restrict manufacturing development and progress, thus narrowing the range of industrial structure upgrading.

In the new macroeconomic environment, there is a relatively aged population and a labor shortage. The acceleration of population aging will further aggravate that labor shortage [1,29]. In recent years, the service industry developed faster than the manufacturing industry. To be exact, the service industry has developed faster than the manufacturing industry since the year 2012. According to a series of reports on economic and social development achievements released by the National Bureau of Statistics of China on 20 September 2022, the added value of China’s service sector increased from 24,485.6 trillion yuan to 60,968 trillion yuan in 2012–2021, with an average annual growth rate of 7.4 percent in 2013–2021 calculated at 1978 constant prices. That is 0.8 percentage points higher than GDP and 1.4 percentage points higher than industrial added value. However, under a constrained labor supply, the labor required to expand the service industry can only absorb more industrial workers. As well, this process is restricted to a particular labor specialty and reduces the speed of labor absorption such that it is far from meeting the expanding demand of the service industry—especially the emerging service industry [30,31,32], which slows down the pace of the service industry development and industrial structure upgrading to some extent. In summary, this paper proposes: 

**Hypothesis** **1.**
*In the new macroeconomic environment, the acceleration of population aging further aggravates the labor shortage, thus narrowing the range of industrial structure upgrading.*


### 2.2. Capital Stock

In the old macroeconomic environment, the overall population aging at an accelerated pace not only means the proportion of the aging population accelerating at this stage, but it also means the baby boomers (especially the baby boom of the 1960s and 1970s, which was the most populous and influential baby boom since the founding of the People’s Republic of China) accelerating into the labor force, the birth rate of which declined obviously under the one-child policy. Therefore, it would accelerate the decline in the population dependency ratio, positively shocking household savings, and strongly supporting investment expansion and capital stock formation. Furthermore, the reduction of labor factors relative to capital is not conducive to the development and expansion of labor-intensive industries at this stage, thus narrowing the range of industrial structure upgrading.

In the new macroeconomic environment, the acceleration of population aging, on the one hand, means the proportion of the elderly population increase at an accelerated pace, and the household savings rate will decline [16,33]. On the other hand, means the absolute size of the working-age labor force, which is already decreasing, will further decline at an accelerated pace, leading to further acceleration of the dependency ratio and negatively impacting household savings and investment [34,35]. Furthermore, due to the diminishing marginal driving effectiveness of capital, the range of industrial structure upgrading will expand instead. Therefore, this paper proposes: 

**Hypothesis** **2.**
*In the new macroeconomic environment, the acceleration of population aging negatively impacts the capital stock, and further expands the range of industrial structure upgrading.*


### 2.3. Technology

In the old macroeconomic environment, on the one hand, as mentioned above, the acceleration of population aging not only lowers the old-age dependency ratio at an accelerated pace and provides relief for young people, which enhances the vitality of the working population and improves the allocation efficiency of labor resources [17,36] but also accelerates improvements in factor endowments, which will stimulate technological innovation [20]. On the other hand, the accelerated aging of the overall population inhibits technological innovation [20]. Furthermore, the impact of technological progress on industrial structure upgrading is restricted by the low efficiency of transfer and transformation of the technological achievements at this stage.

In the new macroeconomic environment, on the one hand, as mentioned above, the acceleration of population aging not only ages the labor force itself but also aggravates the burden on young people [37,38], which directly inhibits the vitality of the labor force and reduces the allocation efficiency of labor resources [17,36], thus not conducive to technological progress and industrial structure upgrading. On the other hand, the accelerated improvement of factor endowments and the accelerated accumulation of educated human capital stimulates technological innovation [18,39,40]. In addition, relying on a massive increase in the labor supply and capital stock to drive the industrial structure upgrading is no longer realistic. Along with the impact of the acceleration of population aging, we can only rely on technological innovation to drive industrial structure upgrading, namely, there is a certain mechanism of reversal pressure on technological innovation [15,19,20,41,42,43,44,45,46,47]. In sum, the net effect on technological progress needs to be further tested and analyzed quantitatively. Furthermore, benefiting from the obvious improvement in the transfer and transformation efficiency of technological achievements at this stage, technological progress has a relatively strong driving force for industrial structure upgrading. Therefore, this paper proposes: 

**Research Hypothesis** **3.**
*In the new macroeconomic environment, the net impact of the acceleration of population aging on technological progress is uncertain; furthermore, technological progress has a relatively strong impact on industrial structure upgrading.*


### 2.4. Consumption

In the old macroeconomic environment, even if a rising elderly population accelerated, the demand for the elderly care service industry would increase slowly in the era of material consumption, and mainly stay at the level of a survival service [48]; durable consumer goods such as household appliances, mobile phones, and clothing are upgraded slowly, and the acceptance and replacement frequency of new products by the elderly is far behind those of the young [49]. In summary, the upgrading range of consumption is relatively small. Furthermore, at this stage the market economic system has not yet been perfected, thus the production and supply is not follow the market demand information [50]. Especially in the field of elderly care services, the upgrading process from welfare to systematic services is mainly guided by government capital [51]; that is, the actual efficiency of industrial structure upgrading driven by the upgrading of consumption is dominated by human factors.

In the new macroeconomic environment, the rising of the elderly population at an accelerated pace generates an aggravatingly expanding demand for elderly care, which includes housekeeping, medical care, and health care [21]. The renewal of durable consumer goods is obviously accelerated, and the acceptance and replacement frequency of new products for the elderly is also increased. The general improvement in living standards makes leisure elderly people’s consumption expenditure on culture, tourism, entertainment, and other aspects increase significantly [22,52]. In short, the upgrading range of consumption is relatively wide. Furthermore, the actual driving efficiency of the upgrading of consumption on industrial structure upgrading is disturbed by many factors, such as the profitability of relevant industrial enterprises, the investment cycle, and the effect of policy support [53]. Therefore, this paper proposes: 

**Research Hypothesis** **4.**
*In the new macroeconomic environment, the acceleration of population aging promotes relatively large consumption upgrading, and the further impact on industrial structure upgrading is uncertain.*


### 2.5. Investment

In the old macroeconomic environment, as mentioned above, the acceleration of population aging leads to a relatively small upgrade in consumption. Furthermore, the efficiency of investment and industrial structure upgrading led by consumption is disturbed by non-market factors.

In the new macroeconomic environment, as mentioned above, the acceleration of population aging leads to a relatively wide upgrade in consumption. Furthermore, the efficiency of investment and industrial structure upgrading guided by consumption is affected by many factors such as the profitability of enterprises in relevant industries, the investment cycle, and the effect of policy support [53], which needs to be tested quantitatively. Therefore, this paper proposes: 

**Research Hypothesis** **5.**
*In the new macroeconomic environment, the impact of the acceleration of population aging on the upgrading of investment and industrial structures is uncertain.*


### 2.6. Exports

In the old macroeconomic environment, as mentioned above, the acceleration of population aging has a relatively small negative impact on the labor supply. However, at this stage, relatively low levels of opening and international trade bring about weak resistance in labor-intensive industries; thus, the upgrading range of exports is narrowed. Furthermore, similar to the situation from the perspective of domestic consumption analyzed above, the market mechanism at this stage is not perfect, and the industrial structure upgrading does not fully respond to external demand signals. Therefore, whether the industrial structure upgrading will follow the narrowing is uncertain, which needs to be tested by subsequent empirical analyses.

In the new macroeconomic environment, the acceleration of population aging further aggravates the labor shortage, thus, the traditional comparative advantage of labor-intensive industries is lost increasingly rapidly [54,55]. Moreover, climbing to higher value-added links and establishing new comparative advantages suffers from “low-end locking”, and being “captured”, mainly because of the interference of global value chain governance in the international division of labor and trade guided by developed countries [56,57,58,59,60], thereby weakening external demand for industrial structure upgrading. Therefore, this paper proposes: 

**Research Hypothesis** **6.**
*In the new macroeconomic environment, the acceleration of population aging leads to the upgrading range of exports narrowing, thus narrowing the industrial structure upgrading range.*


## 3. Materials and Methods

### 3.1. Variables

In order to identify the transition time point of the impacting mechanism of population aging on industrial structure upgrading, relevant time series data covering possible transition time points from 1987 to 2021 were collected and sorted. The proportion of the aged population (*OR*) in China is obtained from the proportion of the population over 65 years old in the total population. The measurement of the proportion of the aged population is usually the proportion of the population over 65 years old or the proportion of the population over 60 years old. In view of the availability and continuity of the data from China, this paper chooses the number of people over 65 years old to calculate the number of the elderly population. Labor Supply (*L*) is measured by the number of working-age people aged 15–64. The calculation of Capital Stock (*K*) The capital stock (K) is the time series based on the year of 1978 calculated by the GDP deflator, which follows the practice of Haojie Shan (2008) [61], and the depreciation rate is 10.96%. Technological Level (*A*) is measured by the number of valid patents. Consumption Level (*C*) is measured by the Engels coefficient, and the lower the index is, the lower the spending on necessities and the higher the consumption of non-necessities will be. Investment Level (*I*) The time series data of fixed asset investment based on the year of 1978 from the *Statistical Yearbook of China’s Fixed Assets Investment* are directly used to calculate *I.* is obtained from the ratio of fixed investment in the secondary industries to fixed social asset investment, and its change has different meanings at different stages; specifically, the ratio rising in the low- and middle-income stages shows the upgrading of investment, whereas the ratio rising at the middle- and high-income stages means the investment structure is deteriorating and vice versa. Exports Level (*X*) The value of export is converted into RMB by the central parity of the exchange rate between the US dollar and RMB in the current year and reduced by the GDP deflator based on the year of 1978. is measured by the ratio of exports of mechanical and electrical products to total exports, and the higher the ratio, the higher the exports level and vice versa. Industrial Structure (*Indus*) The value-added of the tertiary industry used in the calculation of *indus* is reduced by the value-added index of the tertiary industry, while the time series data of GDP based on the year of 1978 is directly taken from *Yearbook of China Statistics.* is measured by the ratio of added value of tertiary industries to GDP, and the higher the ratio, the higher the industrial level and vice versa. The data needed to calculate the above variables are obtained from the official website of the National Bureau of Statistics of China, *Statistical Bulletin of National Economic and Social Development*, *Statistical Yearbook of China’s Fixed Assets Investment*, *Yearbook of China*, *Yearbook of China Statistics*, and *Yearbook of China Demographics* over the study period.

### 3.2. ADF Test

Table 1 shows that the first-order difference series of variables *OR*, *I*, *X* and *Indus*, the logarithmic series of variables *A*, *K* and *L*, and the original levels of variable *C* are all stationary at the 10% significance level. Its economic implications are population aging, investment upgrading, export upgrading, industrial structure upgrading, technological level, capital stock, labor supply, and consumption level. In this paper, the above stationary series are used to construct an empirical model.

### 3.3. Model Specification

Consider a K-dimensional time series vector with lag p-order $y_{t}={\left(y_{1t},\cdots ,y_{kt}\right)}^{\prime },t=1,2,\cdots ,T$. The specification of a normal VAR model is as follows: (1)$$y_{t}=v+A_{1}y_{t-1}+\cdots +A_{p}y_{t-p}+u_{t}$$
where the error term u_t_ is independent and identically distributed, i.e., $u_{t}\sim IID\left(0,\sum \right)$. vis the intercept term. (A_1_, … A_p_) are coefficient matrix of the K-dimensional time series vector.

The normal VAR model assumes that the parameters (A_1_, … A_p_), v, u_t_ are invariable, while the MSVAR model relaxes this assumption, i.e., parameters allowed to be time-varying and dependent on unobservable regime variable $s_{t}\in \left\{1,\ldots ,M\right\}$_,_ which is subject to ergodic and irreducible first-order Markov processes [62,63]. s_t_ is defined by the transition probabilities [62]: (2)$$p_{i,j}=P_{r}\left(s_{t+1}=j|s_{t}=i\right),\;\sum_{j=1}^{M}p_{ij}=1,\;i,j\in \left\{1,\ldots ,\;M\right\}$$

MSVAR model regards the nonlinear data generation process as piecewise linear, while in each regime, the data generation process is linear. Therefore, the specification of an MS-VAR model with intercept term is as follows: (3)$$y_{t}=v\left(s_{t}\right)+A_{1}\left(s_{t}\right)y_{t-1}+\ldots +A_{p}\left(s_{t}\right)y_{t-p}+u_{t}$$

There are 16 kinds of model specifications (not listed due to space limitations. If interested, please request 16 kinds of model specifications from the authors) corresponding to different combinations of whether the parameters ((A_1_, … A_p_), v, µ µ is the (K × 1) dimensional mean of y_t_, where $µ={\left(I_{K}-\sum_{j=1}^{p}A_{j}\right)}^{-1}v$) change depending on regime switching and whether the error term u_t_ has heteroscedasticity [62]. In addition, the number of regimes and the lag order of the model need to be determined.

As shown in Table 2, six groups of MSVAR models are constructed from each environmental perspective according to equation (3) [62], where yt is a three-dimensional column vector, and three endogenous variables including population aging, the element of the macroeconomic environment and industrial structure upgrading affect each other. It is found that, among a variety of setting forms, MSIH (2)-VAR (1) (I and H indicate that the intercept and error variance terms are state-dependent; 2 indicates that the model has two block states; 1 indicates that the model lags by one order) was the best fit and most explanatory with regard to the data. Table 3 shows the specific statistical test results (The table lists the test results under the setting form of MSIH (2)-VAR (1), but does not list the test results if there are different regime numbers and lag orders under the setting form MSIH-VAR. In addition, the test results under the other 15 kinds of model setting forms with different regime numbers and lag orders are not listed. Thus, if interested, please request the data from the authors). According to the significance of the LR test, likelihood value and AIC criterion, the MSIH (2)-VAR (1) model was significantly better than the linear VAR (1) model, which means that there is a significant change in the influencing mechanism along with a change in the environment.

In addition, the maximum likelihood (ML) estimation method was used to determine the model parameters and the probabilities of regime transition (if interested, please request the date from the authors). based on the EM iterative algorithm used in Hamilton (1990) [63]. It should be noted that the EM algorithm makes more stringent assumptions regarding the error term *u*_t_, which is further required to follow a normal distribution, i.e., $u_{t}\sim NID\left(0,\;\Sigma \left(s_{t}\right)\right)$.

## 4. Results and Discussion of Regime Divisions

The model estimation was completed using OX 3.4 software on the GIVEWIN 2.3 platform. Figure 2 shows that irrespective of the aspect of the environment, the influencing mechanism of population aging on industrial structure upgrading shows obvious two-regime characteristics: except for switching regions, the smoothing probability is basically at the level of 1 in each regime. Each regime has a persistence probability above 0.65 Ibid. Specifically, the time node in which the new environment (Regime 2) becomes dominant is slightly different from each perspective: it is 2012 from the perspective of both labor supply and investment structures, 2013 from the perspective of technology, 2015 from the perspective of exports, 2017 from the perspective of capital stock, and 2019 from the perspective of consumption. In conclusion, industrial development comprehensively and deeply transitioned into a new environmental regime in 2019. In a manner of speaking, Regime 1 is called the old environmental regime, and Regime 2 is called the new environmental regime.

It should be noted that from each perspective there are periods of varying lengths since 1988 that fall in the new environmental regime as well. In terms of labor supply, the year 1988 is captured as a clear singularity (Figure 2a), which indicates that labor supply has a sharply increasing shortage here, probably because in the context of an overheating economy, nationwide governance and rectification began in the second half of 1988, which limited production of township enterprises and urban infrastructure, which resulted in many workers returning to the countryside. At the same time, the rise in urban labor-intensive industries generated urgent demand for industrial workers. It is the temporary labor supply and demand mismatch on the eve of the “migrant worker boom” in early 1989 that led to the growing labor shortage. The model specification in this paper can accurately capture this singularity. In terms of capital stock, the period from 1988 to 1991 is also captured (Figure 2b), which indicates that the speed of capital accumulation decreased significantly during this period, probably because of the overheating economy and contractionary fiscal during the second half of 1988 to 1991, which reduced the scale of fixed asset investment. In terms of technology, the captured period is 1988–1989 (Figure 2c), which indicates that technological progress slowed down during that time, probably because technological progress mainly relied on the introduction of foreign capital, and drastic changes in eastern Europe and political and economic turbulence along with domestic political turmoil directly shocked China’s foreign trade business. Thus, the growth rate of FDI decreased significantly.

On the demand side, the period from 1988 to 1995 is captured from the perspective of consumption (Figure 2d), which indicates that consumption level rose significantly during this period. Specifically, urban residents shifted from the traditional “four big items” (the traditional “four big items” referred to radios, sewing machines, watches, and electric fans) to the new “four big items”, (the new “four big items” refers to the refrigerator, washing machine, recorder, and television) and further shifted from “three single one black” to “three double one color”; (from a single door refrigerator to a double door refrigerator, a single cylinder washing machine to a double cylinder washing machine, a single cassette recorder to a double cassette recorder, and a black and white television to a color television) rural residents began to consume the “four big items” and several high-end products. 1988 is captured as a singular point from the perspective of investment (Figure 2e), which indicates that the upgrading of investment accelerated here, probably as a result of cleaning up long-term projects in the general processing industry in a way that resembled “slamming on the brakes”; optimizing investment structure against overheating began in the second half of 1988. Another four obvious singular points are captured from the perspective of investment. First, 1991 was a singular point because three years’ work of cleaning-up, rectification and governance were basically completed at the end of 1991. The growth rate of industrial production under orderly investment resumed as before, and investment level was significantly improved. Second, 1997 was a singular point probably because the central government adopted a moderately tight macro-control policy in response to the economic overheating, which effectively suppressed the real estate boom, thus optimizing investment structure and achieving a balance between supply and demand until the economy successfully achieved a “soft landing” in 1997. Third, 1999 was a singular point. After the “soft landing”, the economy continued to decline until 1999, when the growth rate was only 7.6%. Cyclical overproduction in some industries and poor corporate profits led to tightening investment demand and obvious adjustment and improvement of investment structure. Fourth, 2009 is a singular point probably because the “4 trillion yuan” investment stimulus plan in response to the 2008 financial crisis covered infrastructure including rural infrastructure, self-innovation, health care, and cultural undertakings, which led to an immediate effect and instantly improved investment level. The longest period (1988–2001) is captured from the perspective of exports (Figure 2f), thus suggesting that during this period China’s upgrading of exports was subject to some resistance given that before China’s accession to the WTO at the end of 2001, China was excluded from the multilateral trade and investment framework. On the one hand, the European and American markets are not completely open to China. On the other hand, the liberalization of international trade and investment has not been sufficient for China. In terms of dynamic comparative advantages, China’s exports were shifting from light textile products to mechanical and electrical products at this stage. As an important export body in China, foreign direct investment enterprises from developed countries were restricted in terms of industrial transfer and investment. Therefore, it is difficult to fully support China’s export comparative advantage and upgrading.

## 5. Results and Discussion of Impulse Response Analysis

Based on the cumulative impulse response function and referring to the dynamic influencing mechanism of population aging on industrial structure upgrading in Regime 1, this section focuses on the influencing mechanisms in Regime 2. It should be noted that every positive shock of one standard deviation unit on population aging (D (OR)) implies that the increase in the proportion of the elderly population will expand by one unit; the fluctuations in investment upgrading (DI), export upgrading (DX) and industrial structure upgrading (D (Indus)) below the zero line, respectively, mean that the upgrading ranges of their investment, exports, and industry have narrowed.

### 5.1. Labor Supply

For the MSVAR01 model, it can be seen from Figure 3 and Figure 4 that for a one-unit positive shock on population aging in the current period, the cumulative decline in the labor supply under Regime 2 is more severe (i.e., about four times that under Regime 1). Furthermore, for every one-unit negative shock on the labor supply, the cumulative narrowing of industrial structure upgrading under Regime 2 is relatively small (i.e., about 1/5th of that under Regime 1), and the cumulative narrowing range in 70 years reaches about 8.8 percentage points and has not yet converged. Furthermore, if population aging is positively shocked by one unit in each period, it will have a continuously expanding negative impact on the labor supply and thus on the scale of industrial structure upgrading given the cumulative and superimposed effects.

It can be seen from the above that the acceleration of population aging in the current period directly and negatively shocks the labor supply, which has a long-term impact on industrial structure upgrading. In contrast, in the new environment, the negative impact of the acceleration of population aging on the labor supply is more severe, thus exacerbating the labor shortage. When the total supply is insufficient, the labor needed to expand the service industry will be supplied by industrial workers, who can be directly matched to traditional services, such as the catering and hospitality industries. However, it is difficult to directly match the emerging service industries, such as finance, software and information, which require high-quality labor [30,31,32], thus continuously narrowing the upgrading range of industrial structure.

### 5.2. Capital Stock

For the MSVAR02 model, it can be seen from Figure 5 and Figure 6 that for a one-unit positive shock on population aging in the current period, the cumulative negative response of capital stock under Regime 2 is expanding (i.e., the response under Regime 1 is positive). Furthermore, for every one-unit negative shock on the capital stock, the range of industrial structure upgrading under Regime 2 expands sharply to about 34.25 percentage points at the initial stage, and then slowly and steadily decays to 30.75 percentage points in the 70th year, not yet converged. On this basis, considering the situation that population aging is shocked by one one-unit positive shock in each period, and it will have a strengthening impact on the capital stock under the cumulative and superimposed effect, and thus the impact on the degree of industrial structure upgrading will be steadily strengthened until convergence is achieved.

It can be seen from the above that the acceleration of population aging in the current period will have a long-term impact on the capital stock and industrial structure upgrading. In contrast, in the new environment, if population aging continuously accelerates, it will be difficult for the elderly who only consume and do not produce to support the “high savings-high investment” development model [16,33], thus continuously and negatively impacting the capital stock. Furthermore, the range of industrial structure upgrading expands continuously until convergence as the marginal driving force of capital weakens. 

### 5.3. Technology

For the MSVAR03 model, Figure 7 and Figure 8 show that the technological level responds positively to every one-unit positive shock on population aging in the current period and begins to converge in the 50th period. Moreover, the rise in the technological level under Regime 2 is about 1/3rd of that under Regime 1. Furthermore, every one-unit positive shock on the technological level drives the degree of industrial structure upgrading to expand by about 10% in the current period, then reaches its highest point in the first period—specifically, by about 23.2% in Regime 2, which is far higher than the 14% in Regime 1—then gradually decays and converges to a stable level in the 50th period by about 22 percentage points in Regime 2, which is in sharp contrast to the negative 20 percentage points in Regime 1. On this basis, if population aging in the new environment is affected by a one-unit positive shock in each period, the technological level will continuously improve under the cumulative and superimposed effects, which will further continuously expand the range of industrial structure upgrading.

It can be seen from the above that the impact of the acceleration of population aging in the current period on the technological level and thus on industrial structure upgrading will roughly last for 50 periods and then tend to converge. In contrast, the impact of the acceleration of population aging on technological progress is relatively weak in the new environment [64]. Although there is a certain mechanism for technological progress gradually strengthening [15,19,20,41,42,43,44,45,46,47], on the one hand, along with the decline in labor supply, an aging working population, and higher total dependency ratios, innovation vitality and flow efficiency of the working population obviously decreased, thus impeding technological progress [17,36,38]; on the other hand, China is shifting toward independent innovation at present, and its loss of backward advantage weakens the effect of human capital accumulation on technological progress. Furthermore, the impact of technological progress on industrial structure upgrading is relatively strong, probably because the transfer and transformation mechanism of scientific and technological achievements are obviously improved, and the current technological progress in China is more involved in universality technology and infrastructure industry with a strong positive externalities. Many of them are global pioneers, and such high-quality technological progress has greatly changed our ways of production and life, thereby upgrading the overall industrial levels.

### 5.4. Consumption

For the MSVAR04 model, it can be seen from Figure 9 and Figure 10 that, for every one-unit positive shock of population aging in the current period, the Engels coefficient in Regime 2 cumulatively decreases until it converges in the 50th period, which is about three times the decline amplitude of that in Regime 1. Furthermore, for every one-unit negative shock on the increasement of the Engels coefficient, the range of industrial structure upgrading in Regime 2 cumulatively narrows until it converges in the 50th period, which is in sharp contrast to the cumulative expansion in Regime 1. On this basis, if in the new environment population aging is affected by a one-unit positive shock in each period, consumption levels will continuously improve under the cumulative superposition effect, and the range of industrial structure upgrading will continuously narrow.

It can be seen from the above that the impact of the acceleration of population aging in the current period on consumption and industrial structure upgrading lasts for roughly 50 periods. Compared with the past, the acceleration of population aging in the new environment has a more obvious negative impact on the Engels coefficient; that is, consumption levels increase more significantly. This is most likely due to the size of the elderly population expanding more rapidly based on the more aged population, which generates greater, more diversified, and more personalized demand for elderly care services [21]. Furthermore, the sharp rise in consumption levels fails to drive the expansion of industrial structure upgrading; that is, the range of industrial structure upgrading narrows probably because guided by a large-scale consumption market the pension industry has experienced rapid development, but is subject to an income ceiling (i.e., the revenue ceiling of the elderly population) and a lower cost limit (i.e., nursing staff wages could not be lower than the local minimum wage standard) [65,66,67,68,69], thus the pension industry is still relatively underdeveloped and unable to satisfy market demand [70,71,72]. In terms of scale, the pension industry in developed economies such as Europe and the United States generally accounts for more than 20% of GDP, while the proportion in China is only 7%. China is thus a long way from building a “one-stop shop” pension service center that can cover most of the elderly population in urban and rural areas [73]. Taking pension beds as an example, China should have at least 13 million beds in 2020 to meet expected demand. According to international standards, 5% of the elderly population (i.e., over 60 years old) will enter pension institutions. In fact, there are only 8.075 million pension beds (the data is from the 2020 Statistical Bulletin on the Development of Civil Affairs published on the official website of the Ministry of Civil Affairs), which can only meet about 60% of the needs of the elderly. In terms of its internal structure, it mainly provides essential life services such as basic life care and does not address middle- and high-end and hedonic pension projects.

### 5.5. Investment

For the MSVAR05 model, Figure 11 and Figure 12 show that for every one-unit positive shock of population aging in the current period, the increasing range of the percentage of industrial investment in Regime 2 steadily expands, is slightly higher than that in Regime 1, and begins to level off at about 1.9% from the 15th period. Furthermore, for every one-unit positive shock on the increasing range of industrial investment, the range of industrial structure upgrading cumulatively narrows by about 10 percentage points until the 15th period, then remain at this level. On this basis, if population aging is affected by a one-unit positive shock in each period in the new environment, the degree of investment upgrading will continuously narrow under the cumulative superposition effect, and thus the degree of industrial structure upgrading will continuously narrow.

It can be seen from the above that the impact of the acceleration of population aging in the current period on investment and industrial structure upgrading lasts for about 15 periods. In contrast, in the new environment, the acceleration of population aging brings about a growing elderly population and generates a growing demand for elderly care services, thus attracting investment to this industry. The service target has shifted from the elderly with special difficulties to the elderly in general; the service type has shifted from single pension institutions to integrated forms that include community, home, and institutions [74,75,76]. The investors have shifted from the single government to private capital and the service itself has shifted from the subsistence to the hedonic type [77,78,79]. Investment in the elderly care services industry has seen a massive increase led by consumption, but the upgrading range has narrowed, which suggests that it is still facing obvious bottlenecks, which is consistent with the impulse response analysis from the perspective of consumption. It also suggests that the empirical results of this paper are robust.

### 5.6. Exports

For the MSVAR06 model, it can be seen from Figure 13 and Figure 14 that, for every one-unit positive shock of population aging in the current period, the range of the percentage of exports of mechanical and electrical products under Regime 2 continuously narrows. The cumulative decrease is about 1.2 percentage points until the 7th period, which is about half of that under Regime 1, then remains at this level. Furthermore, with every one-unit negative shock on the range of the percentage of exports of mechanical and electrical products, the range of industrial structure upgrading first expands and then narrows, and the cumulative narrowing reaches about 0.2 percentage points until the seventh period, then remains at this level, which is in sharp contrast to the cumulative expansion of the range of industrial upgrading under Regime 1.

As can be seen from the above, the acceleration of population aging under the new environment leads to the narrowing of the range of the percentage of exports of mechanical and electrical products. That is, the upgrading range of exports narrows, and the narrowing range is about half of that under the old environment, which indicates that exports are less affected by population aging at this stage because the higher-order upgrading of exports itself is relatively slow and the importance of labor-intensive industries is greatly reduced. Specifically, as a pillar industry in China, the electrical and mechanical industry lies in the labor-intensive and low- and medium-end links in the global value chain. Under the new environment, the acceleration of population aging directly leads to obvious rising labor costs and weakens the export comparative advantage of mechanical and electrical products. At the same time, new comparative advantages have not been established, leading to the narrowing of export upgrading. Furthermore, from the external demand side, it drives the range of industrial structure upgrading to narrow, while it has the opposite effect in the old environment, which indicates that the market economic system had not yet been perfected at this stage and the industrial levels had obviously not yet synchronized with the changes in external demand. It should be noted that the narrowing of the upgrading range of exports does not immediately drive the narrowing of the range of industrial structure upgrading; that is, there is a lag effect probably because production adjustment needs a longer time than demand adjustment.

Inside the electromechanical industry, first, the production of optical electronic chips focuses on the middle and low ends. Even though some high-performance chips have achieved independent technological innovation, the high-end market is monopolized by European and American enterprises. Second, in the field of integrated circuit chips there is a sizeable technology gap between European and American countries and China, which mainly occupies the labor-intensive assembly and test links in the global value chain. Under the new environment, the acceleration of population aging weakens comparative advantages. Moreover, in chip design and manufacturing links, China is far from reaching the international level, and it is difficult to establish new comparative advantages in the short run. Furthermore, in the forms of foreign capital and subsidiary domestic capital, China participates in the global production of mobile phones mainly in the processing and assembling links. For example, the largest production base of Apple phones is the Zhengzhou airport area, and inside it, Foxconn is the most representative foundry enterprise; Zhengzhou smartphone production alone accounted for 1/7th of the global share. The data are from the *Government Work Report* in Zhengzhou city in 2021. However, labor-intensive links yield low profits and add little value, which creates great pressure in terms of industrial transfer resulting from rising labor costs and weakening comparative advantages. Finally, in the international division of the automobile industry, China mainly focuses on vehicle and parts manufacturing and exports, and only a small number of self-owned brand automobile enterprises engage in exports based on integrating design, manufacturing, and sales. The acceleration of population aging negatively shocks the export comparative advantage of the automobile industry in its labor-intensive links.

## 6. Conclusions and Discussion

In recent years, the macroeconomic environment has changed comprehensively and profoundly, and population aging, as an important context for industrial and economic development, is continuing to impact and shape the macroeconomic environment. There are obvious differences in the influencing mechanisms of population aging on industrial structure upgrading under different environmental constraints. In this paper, six groups of MSVAR models are constructed from the perspective of each element of the macroeconomic environment. The regime transition probability figures show that the influencing mechanism of population aging on industrial structure upgrading has obvious two-regime characteristics from any perspective and was comprehensively transferred to the new economic regime in 2019. Furthermore, compared with the situation under the old environment, under the new environment the acceleration of population aging aggravates the labor shortage, thus narrowing the scale of industrial structure upgrading; negatively rather than positively impacts the capital stock, however, the range of industrial structure upgrading widens due to the decreasing marginal impact of capital; has a weaker effect on technological progress, but further leads to a stronger impact on industrial structure upgrading, which probably because the transfer and transformation efficiency of technological achievements is obviously improved; better improves consumption levels, however, it weakens industrial structure upgrading; narrows rather than widens the scale of investment and industrial structure upgrading; narrows the scale of exports and industrial structure upgrading to a lesser extent. It should be noted that population aging has sustained long-term impacts on the labor supply and capital stock, about 50 periods of impact on the technological and consumption levels, about 15 periods of impact on investment upgrading, and about seven periods of impact on export upgrading. If we consider the cumulative and superimposed effects of the continuous acceleration of population aging, the impact on industrial structure upgrading will be more profound.

In view of the omnidirectional and systematic impact of population aging on the macro environment of industrial development, a country should collaboratively promote the upgrading of industrial structure at both the supply and demand sides. Industrial structure upgrading. On the supply side, innovation should always be regarded as the first driver of industrial structure upgrading. Relying on large-scale inputs of labor, capital, and other production factors to drive industrial structure upgrading is only temporary, finally, it should be replaced by technological progress to drive, thus we must rely heavily on innovation. The nature of innovation-driven is talent-driven. On the one hand, a country should continuously optimize its innovative talents training mechanism, to build a backbone force for the transmission of innovative talents; on the other hand, the country should continuously improve its hard environment (such as updating its outdated infrastructure) and soft environment (such as relaxing its skilled immigration policy) to strengthen its talent attraction. In addition, latecomer countries should attach importance to and strengthen the transformation of technological achievements and further smooth the driving mechanism of technological progress on industrial structure upgrading by developing and expanding technology trading service institutions, introducing professional talent, increasing support for special funds.

Industrial structure upgrading industrial structure upgrading industrial structure upgrading on the internal demand side, we should rely on the upgrading of domestic consumption level to drive industrial structure upgrading. On the one hand, a country should fully tap and release the consumer market potential of the elderly population. With the continuous expansion of the scale of the elderly population, the potential consumption market of the elderly population continues to expand. In most developed countries, the elderly consumption proportion has already reached more than half, and the income of the elderly population is relatively high, so the release of their consumption potential is not subject to income constraints. The key point is to constantly develop products and services to meet the needs of the elderly products and services according to the market gap. On the other hand, the latecomers should smooth the driving mechanism of consumption level upgrading on the investment and industrial structure upgrading. Especially, they should break the income ceiling of pension service enterprises to break through their development bottleneck and increase their profitability. First, it is necessary to increase the income of the elderly. Steadily increase the pension standard and implement policies to delay retirement and encourage reemployment after retirement. Second, provide certain subsidies for the elderly necessities, so that the elderly can have more surplus income for the consumption of pension services. Third, we should improve the medical insurance system to provide relief to the elderly.

On the external demand side, industrial structure upgrading should be driven by the upgrading of the exports level. Population aging continuously weakens the comparative advantages of labor-intensive products and links and hinders the upgrading of the production and exports level of emerging countries such as India, Mexico, Brazil and Russia, as well as Southeast Asian countries such as Indonesia and Thailand. In addition, given the overall sluggish world economy and sluggish external demand, it is more important to further open up, strengthen international cooperation, and increase the introduction of high-quality capital and enterprises. At the same time, local enterprises with certain scales and strength can extend their industrial chain through mergers and acquisitions, and gradually realize diversified production and level improvement of products.

The limitation of this study and future research directions lie in the following aspects. First, our research mainly focuses on the influence mechanism of population aging on industrial structure upgrading through empirical analysis, and lacks strict theoretical modeling and mathematical derivation and demonstration. In the future, a multi-sector growth model can be established, and the production sector can be subdivided into agriculture, industry, and services. At the same time, the OLG model based on the life cycle theory can be used to describe the intergenerational relationship of consumer behavior. Second, this study made empirical analysis with the empirical data at the national level, and did not consider the regional diversity of the impact of aging within a country. Given the work to deal with population aging is complicated, we can compare the influence mechanism of population aging on the upgrading of industrial structure at the provincial level in the future. Third, the policy implications of responding to population aging for a country have their limitations, given they are based on the experience of China. Although there are similarities among countries in the world in terms of population aging, there are still great differences. In the future, heterogeneity can be fully considered, and comparative analysis can be conducted based on cross-country data.

## Figures and Tables

**Figure 1 ijerph-19-16093-f001:**
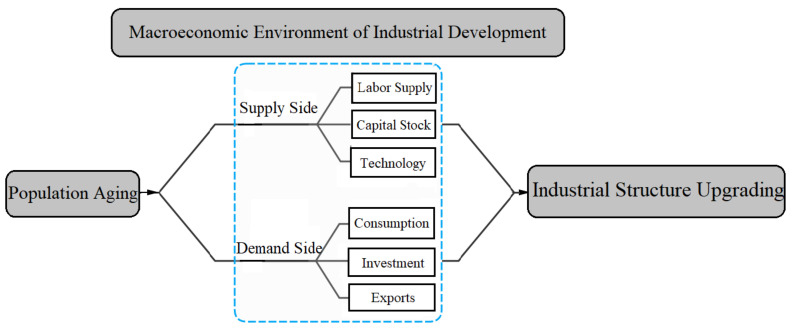
The Influencing Mechanism of Population Aging on Industrial structure upgrading.

**Figure 2 ijerph-19-16093-f002:**
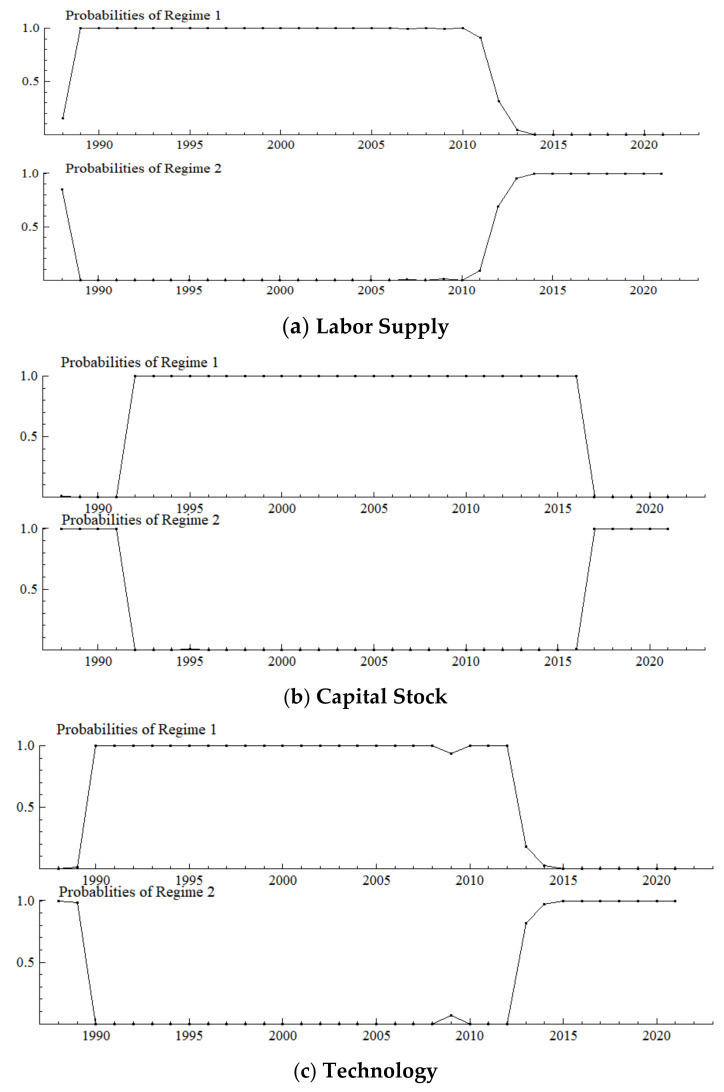
Probabilities of a Smooth Regime Transition.

**Figure 3 ijerph-19-16093-f003:**
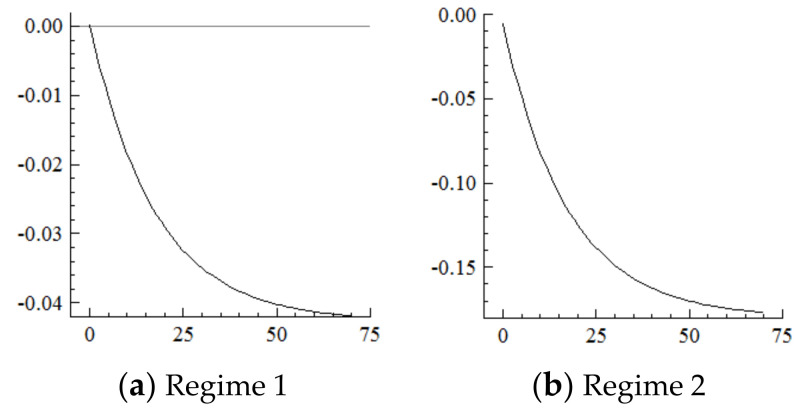
Cumulative Impulse Response of Labor Supply to Population Aging Shocks.

**Figure 4 ijerph-19-16093-f004:**
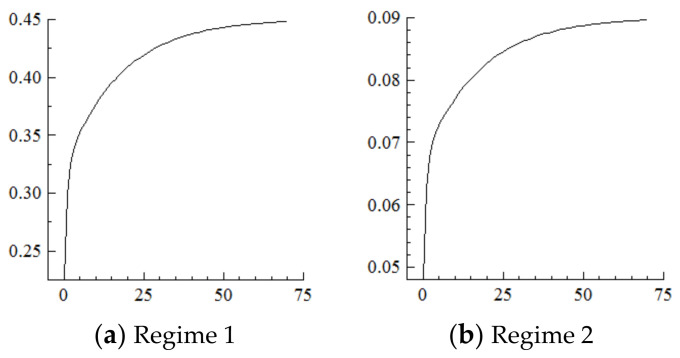
Cumulative Impulse Response of Industrial structure upgrading to Labor Supply Shocks.

**Figure 5 ijerph-19-16093-f005:**
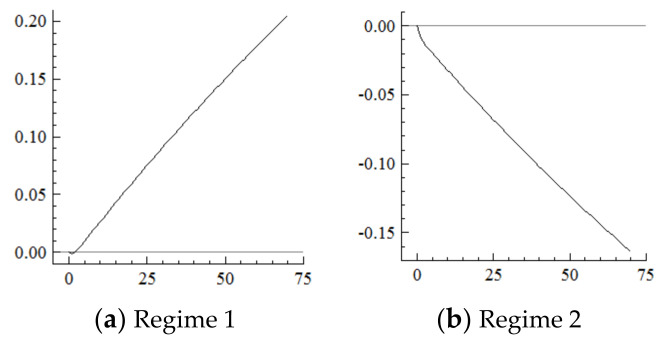
Cumulative Impulse Response of the Capital Stock to Population Aging Shocks.

**Figure 6 ijerph-19-16093-f006:**
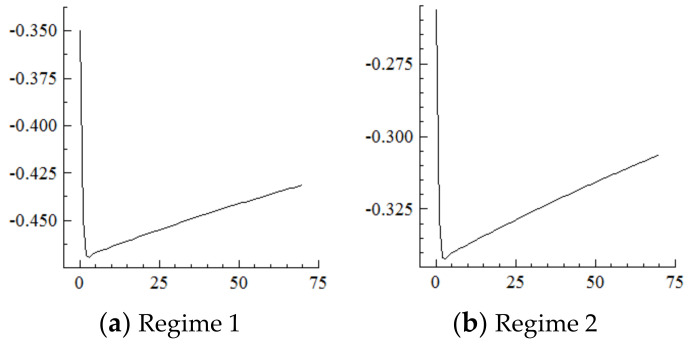
Cumulative Impulse Response of Industrial Structure Upgrading to Capital Stock Shocks.

**Figure 7 ijerph-19-16093-f007:**
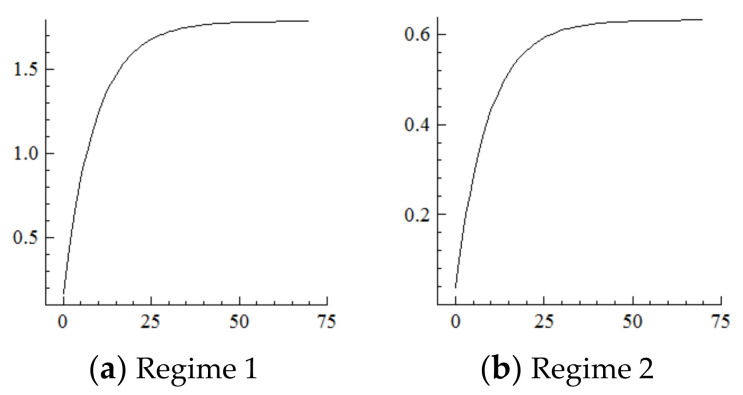
Cumulative Impulse Response of the Technological Level to Population Aging Shocks.

**Figure 8 ijerph-19-16093-f008:**
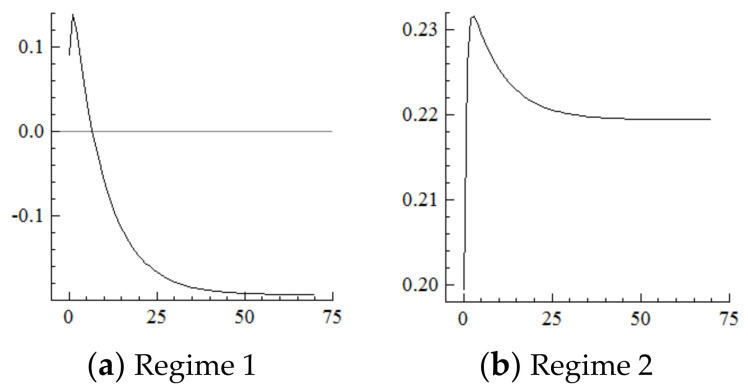
Cumulative Impulse Response of Industrial structure upgrading to Technological Shocks.

**Figure 9 ijerph-19-16093-f009:**
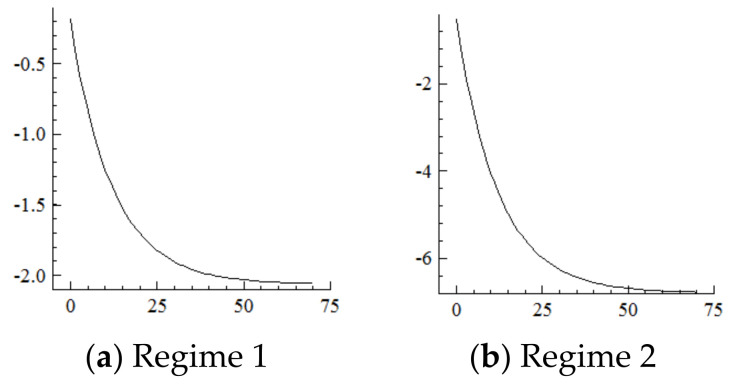
Cumulative Impulse Response of Consumption to Population Aging Shocks.

**Figure 10 ijerph-19-16093-f010:**
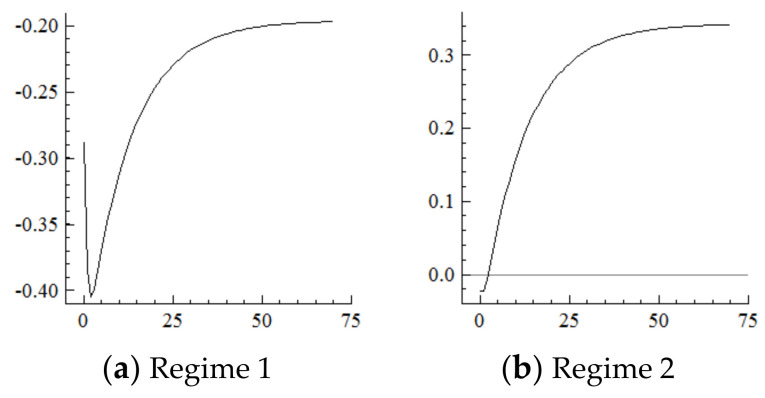
Cumulative Impulse Response of Industrial Structure Upgrading Industrial structure upgrading to Consumption Shocks.

**Figure 11 ijerph-19-16093-f011:**
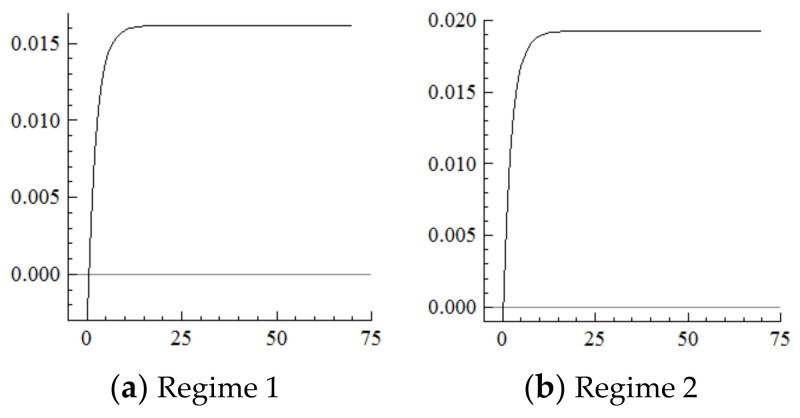
Cumulative Impulse Response of Investment to Population Aging Shocks.

**Figure 12 ijerph-19-16093-f012:**
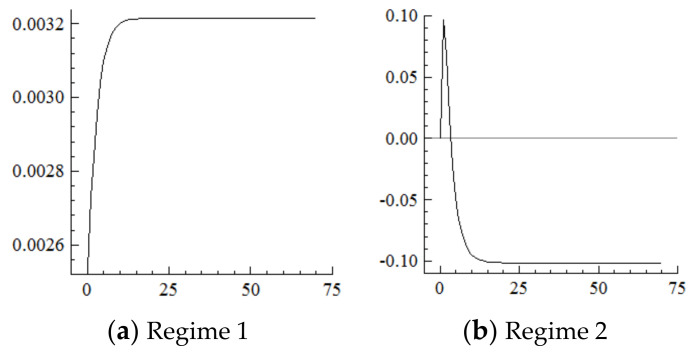
Cumulative Impulse Response of Industrial structure upgrading to Investment Shocks.

**Figure 13 ijerph-19-16093-f013:**
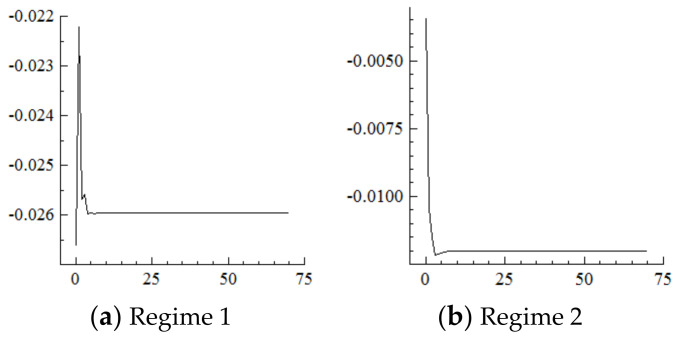
Cumulative Impulse Response of Export Upgrading to Population Aging Shocks.

**Figure 14 ijerph-19-16093-f014:**
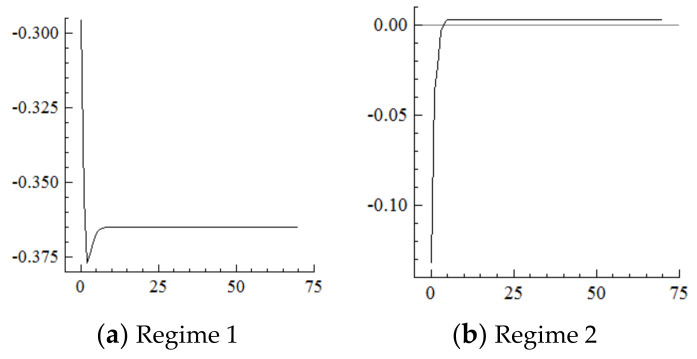
Cumulative Impulse Response of Industrial Structure Upgrading to Export Upgrading Shocks.

**Table 1 ijerph-19-16093-t001:** Stationary test of each time series.

Variable Form	Test Form (C, T, K)	ADF Statistics
D(or)	(C, 1, 0)	−3.5866 ** (0.0461)
lnL	(0, 0, 5)	−1.7575 * (0.0749)
lnK	(C, 0, 2)	−3.2342 ** (0.0271)
lnA	(C, 1, 0)	−5.4977 *** (0.0004)
C	(0, 0, 0)	−3.5169 *** (0.0009)
DI	(0, 0, 0)	−4.5599 *** (0.0000)
DX	(0, 0, 0)	−5.3933 *** (0.0000)
D(indus)	(0, 0 0)	−3.0119 *** (0.0037)

Note: Each item in test form (*C*, *T*, *K*) represents a constant term, trend term and lag term, respectively. The lag order is determined according to the SIC criterion. *, ** and *** indicate that the null hypothesis of the existence of unit root in time series is rejected at the 1%, 5% and 10% levels, respectively.

**Table 2 ijerph-19-16093-t002:** MSVAR Model Specifications.

Model Name	Each Element of the Macroeconomic Environment	Model Variables	Model Specifications
MSVAR01	Labor Supply	D(or), InL, D(indus)	MSIH (2)-VAR (1)
MSVAR02	Capital Stock	D(or), InK, D(indus)	
MSVAR03	Technological Level	D(or), InA, D(indus)	
MSVAR04	Consumption Level	D(or), C, D(indus)	
MSVAR05	Investment Level	D(or), DI, D(indus)	
MSVAR06	Exports Level	D(or), DX, D(indus)	

**Table 3 ijerph-19-16093-t003:** MSVAR Model Test Results.

Model Name	LR Test Statistic	Log Likelihood	AIC/SC Criterion Test
MSVAR01	LR linearity Test: 29.5975 DAVIES = [0.0124] * Chi (9) = [0.0005] **, Chi (11) = [0.0018] **	MSVAR Model: 244.8354 VAR Model: 230.0366	MSVAR Model: −12.6962/−11.6647 VAR Model: −12.4727/−11.3943
MSVAR02	LR linearity Test: 33.4084 DAVIES = [0.0032] ** Chi (9) = [0.0001] **, Chi (11) = [0.0005] **	MSVAR Model: 235.8265 VAR Model: 219.1223	MSVAR Model: −12.1663/−11.0226 VAR Model: −11.8307/−10.8644
MSVAR03	LR linearity Test: 61.4635 DAVIES = [0.0000] ** Chi (9) = [0.0000] **, Chi (11) = [0.0000] **	MSVAR Model: 139.2030 VAR Model: 108.4713	MSVAR Model: −6.4825/−5.1806 VAR Model: −5.3218/−4.5138
MSVAR04	LR linearity Test: 32.7775 DAVIES = [0.0040] ** Chi (9) = [0.0001] **, Chi (11) = [0.0006] **	MSVAR Model: 95.6081 VAR Model: 79.2194	MSVAR Model: −3.9181/−2.7931 VAR Model: −3.6011/−2.6162
MSVAR05	LR linearity Test: 40.0504 DAVIES = [0.0003] ** Chi (9) = [0.0000] **, Chi (11) = [0.0000] **	MSVAR Model: 232.6291 VAR Model: 212.6039	MSVAR Model: −11.9782/−10.6763 VAR Model: −11.4473/−10.6392
MSVAR06	LR linearity Test: 44.0805 DAVIES = [0.0001] ** Chi (9) = [0.0000] **, Chi (11) = [0.0000] **	MSVAR Model: 230.1236 VAR Model: 208.0834	MSVAR Model: −11.8308/−10.5289 VAR Model: −11.1814/−10.3733

Note: * and ** indicate that the null hypothesis of the linearity of model is rejected at the 1% and 5% levels, respectively.

## Data Availability

Not applicable.

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
