# Peer review of "How Population Aging Affects Industrial Structure Upgrading: Evidence from China"

_ijerph, 2022, doi:10.3390/ijerph192316093_

Round 1
Reviewer 1 Report
The studied theme is very interesting because focuses on how population aging affects industrial upgrading under the environmental constraints. The paper put six theoretical hypotheses from the perspective of each element of the macroeconomic environment (e.g., labor supply, capital supply, technology, consumption, investment and exports). Then, a time-varying MSVAR model is used to determine the transition point in the influencing mechanism of population aging on industrial upgrading, as well as that of the environment, further to lay special stress on analyzing the influencing mechanism of population aging on industrial upgrading in the context of the new macroeconomic environment.
Major comments
1. Literature review is adequate to the subject. The novelty of research is clearly presented. Research hypotheses were clearly stated, and the results were correlated with them.
2.The methodology is accurately described regarding the econometric models used in the paper. The results are detailed described.
3.The language is very clear and readable.
4.I only suggest future research directions may also be highlighted.
Author Response
Dear expert, thank you very much for your affirmation of the writing value of this paper. Based on your comments and inspiration we have made the following change.
Reviewer 1 Comments:
- I only suggest future research directions may also be highlighted.
Response: At the end of the paper, we added a discussion of research limitations and future research directions.
Reviewer 2 Report
The article describes the impact of population aging on the modernization of industry under environmental constraints on the example of China. The paper focuses on the research on the influencing mechanism of population aging on industrial upgrading under the environmental constraints, and further constitutes synergy mechanism from the supply and demand sides in adapting to changes in the macroeconomic environment to promote industrial upgrading while in market equilibrium. Many research questions have been formulated in the article. To answer these questions, the paper seeks to put forward six theoretical hypotheses from the perspective of each element of the macroeconomic environment.
The article is mainly empirical. The research questions are correctly formulated. The research methods were selected correctly. The results of the analysis are presented clearly. The conclusions are consistent, they result from the conducted research. The selection of source materials is adequate and up-to-date. The article has the correct structure. The figures are legible and help to better understand the analyzed issues.
Summing up, it should be stated that the article is coherent and logical. In my opinion, the article is interesting and points to specific solutions to the problem under study.
Author Response
Dear expert, Thank you very much for your detailed reading and positive comments.
Reviewer 3 Report
This is an esoteric paper about the effect of aging on industrial development with environmentally conscious metrics. Data is from China, which makes the outcomes of the study solid and benchmark. Topic is suitable for IJERPH. Paper is very well written and organized. Good literature review is provided.
I only have the following minor recommendations to authors.
- I think that the abstract does not reflect the context clearly. I strongly advise a revision on abstract. It is written more like in a conclusion format. Inclusion of topic’s importance with highlights is crucial.
- Some of the figure legends are misplaced and miscentered.
- Some symbolic text is fuzzy. Line 333, line 345. Please write them in Math editor.
Author Response
Dear expert, thank you very much for your affirmation of the writing value of this paper. Based on your comments and inspiration we have made the following change.
Reviewer 3 Comments:
- I think that the abstract does not reflect the context clearly. I strongly advise a revision on abstra-ct. It is written more like in a conclusion format. Inclusion of topic’s importance with highlights is crucial.
Response: We made a revision on abstract, and the highlights of the topic’s importance is added.
(2) Some of the figure legends are misplaced and miscentered.
Response: We recentered the headings of figure 4, 9, and 10, rearranged the headings of figure 11 to the correct position, and rearranged figure 12 to the correct position.
(3) Some symbolic text is fuzzy. Line 333, line 345. Please write them in Math editor.
Response: We rewrote the symbolic text that you mentioned in Math editor, which is in the line 359 and 390 of the revised version now.
Author Response
Dear expert, thank you very much for your time involved in reviewing the manuscript, the revisions have been made one by one according to your valuable suggestions.
Reviewer 4 Comments:
(1) When I see the title of the paper, I think the word“environmental constraints” means the meaning of resources and environmental constraints, in this paper, it means the macroeconomics environmental in fact. Which shocked me actually, the paper used the word “shock” many times.
Response: We quite accept your point of view, and removed "environmental constraints" from the title of the paper to avoid a mismatch between the title and the text.
(2) Please explain the difference or the time nodes of old macroeconomic environment(L141 etc) and the new macroeconomic environment(L148 etc), “in recent years(L150). except that, the key factors such as capital stock, technology, consumption, investment ,and exports used in this paper, they are preconditions or need to be discerned. It seems they were confusion. Especially the “macroeconomic environment”, sometimes it is the cause, sometimes it is the effect.
Response: The difference between the old macroeconomic environment and the new macroeconomic environment is briefly described in the first paragraph of section 2 “Research hypothesis and theoretical analysis”, i.e.,“In the past, the macroeconomic environment of industrial development was characterized by an abundant labor supply, rapid capital accumulation, rapid technological progress via imitative innovation, slow upgrading of consumption in the declining stage of the “U-shape,” slow investment upgrading, and rapid upgrading of low-order exports given the factor endowments and market power. In recent years, the macroeconomic environment has undergone fundamental changes, which have been characterized by a short labor supply, slowed capital accumulation, slowed technological progress resorting to independent innovation, rapid upgrading of consumption in the rising stage of the “U-shaped” curve, accelerated optimizing investment structure in the development stage of emphasizing on quality, and slowed upgrading of higher-order exports restricted by more non-market factors.”(L139-L149) Concretely, from the perspective of 2.1 "labor supply", the difference between the old macroeconomic environment and the new macroeconomic environment is as described in the text, "In the old macroeconomic environment, there was a relatively young population and an almost-infinite labor force. "(L161-162) and" In the new macroeconomic environment, there is a relatively aged population and a labor shortage."( L169-170)
The time node of the old macroeconomic environment and the new macroeconomic environment is tested in section 4 "Results and Discussion of Regime Divisions", " it is 2012 from the perspective of both labor supply and investment, 2013 from the perspective of technology, 2015 from the perspective of exports, 2017 from the perspective of capital stock, and 2019 from the perspective of consumption. In conclusion, industrial development comprehensively and deeply transitioned into a new environmental regime in 2019. "(L397-401)
"In recent years, the service industry developed faster than the manufacturing industry"(L171-172). From the statistical data, the time node is about 2012. According to a series of reports on economic and social development achievements released by the National Bureau of Statistics of China on September 20th, 2022, the added value of China's service sector increased from 24,485.6 trillion yuan to 60,968 trillion yuan in 2012-2021, with an average annual growth rate of 7.4 percent in 2013-2021 calculated at 1978 constant prices. That is 0.8 percentage points higher than GDP and 1.4 percentage points higher than industrial added value. The statistical description has been added to the footnote 3(L172).
In fact, Capital stock, technology, consumption, investment, export and other key elements of macroeconomic environment used in this paper are both threshold variables (the environment is the "cause" ) and intermediate variables (the environment is the "result" sufferring from aging shock). Specifically, the influence of population aging on industrial structure is nonlinear, and environmental change is the cause of such nonlinearity. With the change of environment, the influence mechanism of population aging on industrial structure upgrading will also change, that is, the influence mechanism of population aging on industrial structure upgrading under the old environment is different from that under the new environment(In section 2 theoretical analysis and section 5 impulse response analysis, from each perspective, the influence mechanism of population aging on industrial structure upgrading under the old environmental constraints is taken as a contrast, and the influence mechanism of population aging on industrial structure upgrading under the new environmental constraints is mainly analyzed.), the econometric model does test the obvious two-regime future (section 4), specifically identifying the time node from the old environment to the new environment is the same as I answered you in the previous paragraph. At the same time, the aging of population affects the upgrading of industrial structure through the macroeconomic environment (figure 1), in which the macroeconomic environment is the intermediate variable, and the aging of population will have an impact on all aspects of the macroeconomic environment.
(3) The data used in this paper should be processed properly for example according to CPI or something else based on the key factors.
Response: I'm so sorry that the translation of this economic variable is inaccurate and inconsistent. In fact, this indicator is sometimes called “industrial upgrading” and sometimes called “industrial structure upgrading”(e.g.,L36, L273, L580) in this paper. More importantly, What this paper has been discussing is the change, optimization and upgrading of industrial structure. Concretely, It mainly involves the development status of manufacturing industry (light textile industry, mechanical and electrical industry), service industry (traditional service industry, emerging service industry) at different stages and in different environments, as well as the change of these industries’ importance in economic development. In this regard, the name of the economic variable has been unified in the title and the full text as “industrial structure upgrading”. In this case, it is appropriate that the indicator “industrial structure (indus)” is measured by the ratio of added value of tertiary industries to GDP.
(4) The data used in this paper should be processed properly for example according to CPI or something else based on the key factors.
Response: In fact, the data used in this article have already been processed or taken directly from the yearbook with price factors removed. The variables involved in price factors in this paper include capital stock (K), investment (I), export (X) and industrial structure (indus). First, the capital stock (K) is the time series based on the year of 1978 calculated by the GDP deflator. Second, the time series data of fixed asset investment based on the year of 1978 from Statistical Yearbook of China’s Fixed Assets Investment are directly used to calculate investment (I). Third, the amount value of export is converted into RMB by the central parity of the exchange rate between US dollar and RMB in the current year and reduced by the GDP deflator based on the year of 1978. Fourth, The value-added of the tertiary industry used in the calculation of industrial structure (indus) is reduced by the value-added index of the tertiary industry, while the time series data of GDP based on the year of 1978 is directly taken from Yearbook of China Statistics . Accordingly, some footnotes have been added in Section 3.1.
(5) It is necessary to explain the model MSVAR, and the difference between MSVAR AND VAR,
and how to set up the Regime transfer model,the parameters should be explained also, and LR tested should be reported, based on the AIC, SC results, to choose the proper model.
Response: The relevant content has been added in section 3.3 according to your requirements. To be specific, we started from setting up an ordinary linear VAR model, and the meaning of each parameter was explained. Take the VAR model as a reference, when the parameters are variable depends on the change of regime state which is subject to the discrete Markov random process, it becomes a nonlinear MSVAR model, and the equation of setting up a MSVAR model was given. In addition, the more complete results of LR test and the SC criterion test results of the lag order was added in Table 3.
(6) Some details should be improved, especially in the description of the key factors, if the paper can prove its point in data, L143-L144, “in the past, the manufacturing industry developed faster than the service industry”, if there are descriptive statistics for these factors, that is would be better.
Response: According to the annual industrial added value and service added value released on the official website of the National Bureau of Statistics in China, we respectively calculated their average annual growth rate. The average annual growth rate of industrial added value from 1981 to 2011 reached 11.5% (we have deflated the data of industrial added value with the industrial added value index based on the year of 1978), much higher than the average annual growth rate of service added value of 6.12% (we have deflated the data of service added value with the service added value index based on the year of 1978). The statistical description has been added to the footnote 2 in the article(L164).
Reviewer 5 Report
1) maybe the term MSVAR model is clear to statisticians and mathematicians, but the full model name could make the text clearer to other readers.
2) "Based on data from 1987 to 2021, six MSVAR models are constructed from each environmental perspective." missing references?
3) Population is one of the most important factors affecting industrial upgrading [8,9]. - missing space.
4) As authors deal with the problem of aging - additional references to the silver economy are an important issue.
5) "The proportion of the aged population 273 (OR) is obtained from the proportion of the population over 65 years old in the total population."- Is this proportion the Worldwide one, US one? How it was selected?
6) "Consumption Level(C)" - missing space
7) line 323 - formulas look like they are images, not proper equations.
8) line 333 - math formulas and text are mixed so it's hard to read and understand that part of the paper.
9) line 345 - the equation is separated from the text.
10) More general outcomes are missing (if authors can see such outcomes). Otherwise is a China-related example and we don't know if it can be seen as a more general rule for other economies.
11) What type of software was used in the calculations?
Author Response
Dear expert, thanks you very much for your detailed review of this article. In response to your comments, our reply is as follows.
Reviewer 5 Comments:
(1) maybe the term MSVAR model is clear to statisticians and mathematicians, but the full model name could make the text clearer to other readers.
Response: The full name "Markov-Switching Vector Autoregressions" was added when MSVAR appeared for the first time in the text (line111).
- "Based on data from 1987 to 2021, six MSVAR models are constructed from each environmental perspective." missing references?
Response: The reference about the econometric approach to model construction has been added.(line 373-374).
- Population is one of the most important factors affecting industrial upgrading [8,9]. - missing space.
Response: Space has been added.(line 73).
(4) As authors deal with the problem of aging - additional references to the silver economy are an important issue.
Response: We quite accept your point of view. The silver economy in our paper mainly involves two aspects: the silver industry and the silver consumption. The relevant contents mainly appear in the theoretical analysis(section 2.4, 2.5) and the empirical analysis (section 5.4, 5.5). However, There are only six corresponding references([21]-[25],[47]). Therefore, 10 corresponding international references have been added on this issue corresponding to the relevant sections.
(5) "The proportion of the aged population 273 (OR) is obtained from the proportion of the population over 65 years old in the total population."- Is this proportion the Worldwide one, US one? How it was selected?
Response: We are so sorry that we didn't write it clearly here, actually, we are referring to the proportion of the elderly population in China. The measurement of the proportion of the aged population is usually the proportion of the population over 65 years old or the proportion of the population over 60 years old. In view of the availability and continuity of the data from China, this paper chooses the number of people over 65 years old to calculate the number of the elderly population. Some instructions(line 301) and explanations(footnote 5) has been added in the text.
(6) "Consumption Level(C)" - missing space
Response: Space has been added (line 305).
- line 323 - formulas look like they are images, not proper equations.
Response: It has been re-written in Math editor(line 354).
- line 333 - math formulas and text are mixed so it's hard to read and understand that part of the paper.
Response:The math formulas has been re-written in Math editor and placed in the correct position(line 359).
- line 345 - the equation is separated from the text.
Response: The equation has been re-written in Math editor and placed in the correct position(line 390).
- More general outcomes are missing (if authors can see such outcomes). Otherwise is a China-related example and we don't know if it can be seen as a more general rule for other economies.
Response: In fact, China's experience is of great reference significance to other economies. In section 6, the outcomes and implications have been raised to a more general level, which can be seen as a more general rule for both China and other economies.
- What type of software was used in the calculations?
Response: To be more precisely, both the empirical analysis in part 4 and part 5 were completed using OX 3.4 software on the GIVEWIN 2.3 platform. The version description of the software has been added (Line392).
Round 2
Reviewer 4 Report
Comments on “How Population Aging Affects Industrial Structure Upgrading: Evidence from China”
The authors have made a great improvement, and my opinions have been considered accordingly, I am appreciate for that.
And I think there is one paper need to be referred.
DOI:10.19581/j.cnki.ciejournal.2015.11.004
But I do think “population Aging” itself is one of the Macroeconomic environmental”. As for the “Influencing Mechanism of Population Aging on Industrial structure upgrading”(figure 1), the paper I mentioned above need to be read carefully, which explained the mechanism very clearly.
Thank you very much.
Author Response
The paper has been referred.
Reviewer 5 Report
I think that the changes make this paper much better and thus it can be published in the present form.
Author Response
Thanks